

# Evaluation of computational methods for human microbiome analysis using simulated data

Matthieu J. Miossec[1], Sandro L. Valenzuela[1], Marcos Pérez-Losada[2], W. Evan Johnson[3,4], Keith A. Crandall[2] and Eduardo Castro-Nallar[1,2]

[1] Center for Bioinformatics and Integrative Biology, Facultad de Ciencias de la Vida, Universidad Andres Bello, Santiago, Chile
[2] Computational Biology Institute and Department of Biostatistics and Bioinformatics, Milken Institute School of Public Health, The George Washington University, Washington, DC, USA
[3] Section of Computational Biomedicine, Department of Medicine, Boston University, Boston, MA, USA
[4] Bioinformatics Program, Boston University, Boston, MA, USA

Corresponding author
Eduardo Castro-Nallar,
eduardo.castro@unab.cl

## ABSTRACT

**Background:** Our understanding of the composition, function, and health implications of human microbiota has been advanced by high-throughput sequencing and the development of new genomic analyses. However, trade-offs among alternative strategies for the acquisition and analysis of sequence data remain understudied.

**Methods:** We assessed eight popular taxonomic profiling pipelines; MetaPhlAn2, metaMix, PathoScope 2.0, Sigma, Kraken, ConStrains, Centrifuge and Taxator-tk, against a battery of metagenomic datasets simulated from real data. The metagenomic datasets were modeled on 426 complete or permanent draft genomes stored in the Human Oral Microbiome Database and were designed to simulate various experimental conditions, both in the design of a putative experiment; read length (75–1,000 bp reads), sequence depth (100K–10M), and in metagenomic composition; number of species present (10, 100, 426), species distribution. The sensitivity and specificity of each of the pipelines under various scenarios were measured. We also estimated the relative root mean square error and average relative error to assess the abundance estimates produced by different methods. Additional datasets were generated for five of the pipelines to simulate the presence within a metagenome of an unreferenced species, closely related to other referenced species. Additional datasets were also generated in order to measure computational time on datasets of ever-increasing sequencing depth (up to $6 \times 10^7$).

**Results:** Testing of eight pipelines against 144 simulated metagenomic datasets initially produced 1,104 discrete results. Pipelines using a marker gene strategy; MetaPhlAn2 and ConStrains, were overall less sensitive, than other pipelines; with the notable exception of Taxator-tk. This difference in sensitivity was largely made up in terms of runtime, significantly lower than more sensitive pipelines that rely on whole-genome alignments such as PathoScope2.0. However, pipelines that used strategies to speed-up alignment between genomic references and metagenomic reads, such as kmerization, were able to combine both high sensitivity and low run time, as is the case with Kraken and Centrifuge. Absent species genomes in the database mostly led to assignment of reads to the most closely related species

available in all pipelines. Our results therefore suggest that taxonomic profilers that use kmerization have largely superseded those that use gene markers, coupling low run times with high sensitivity and specificity. Taxonomic profilers using more time-consuming read reassignment, such as PathoScope 2.0, provided the most sensitive profiles under common metagenomic sequencing scenarios. All the results described and discussed in this paper can be visualized using the dedicated R Shiny application (https://github.com/microgenomics/HumanMicrobiomeAnalysis). All of our datasets, pipelines and results are made available through the GitHub repository for future benchmarking.

## INTRODUCTION

Metagenomics is emerging as the highest-resolution approach to study the human microbiome, providing essential insights into human health (*Belizário & Napolitano, 2015*). High-throughput sequencing techniques have unleashed vast amounts of metagenomic data, which in turn have prompted the rapid development of sophisticated bioinformatic tools and computational pipelines. A subset of these tools and pipelines are dedicated to providing an answer to the quintessential question "who is there?" (*Grice & Segre, 2012*). By using shotgun metagenome sequencing data and aligning reads against reference databases, we can answer this question at various levels of interest. With taxonomic binning, individual sequence reads are clustered into new or existing operational taxonomic units, obtained through sequence similarity and other intrinsic shared features present in reads. With taxonomic profiling the focus is on estimating the presence and quantity of taxa in a microbial population as well as the relative abundance of each species present. The taxonomic profiling of several metagenomes can in turn give us an understanding of the alpha diversity of a given microbiome across several individuals.

The fact that most microbial species, what *Rappé & Giovanoni (2003)* categorize as the "uncultured microbial majority", cannot be grown in laboratory conditions poses a real challenge for taxonomic profiling. Given the extreme richness of certain communities, such as those residing in soil or sea/freshwater, we can anticipate that a strong proportion of microorganisms extracted and sequenced from those environments will be entirely novel, precluding any taxonomic profiling. However, a substantial fraction of the diversity of microbial communities living within the human body can and has been captured through large-scale studies, such as the Human Microbiome Project (HMP) in the United States or the metagenomics of the human intestinal tract (MetaHIT) project in Europe (*The Human Microbiome Project, 2012*; *Ehrlich & MetaHIT Consortium, 2011*). To support its primary mission of deriving a core human microbiome using metagenomics, the HMP built a reference database by performing whole-genome sequencing and de novo genome-assembly on thousands of bacteria isolated from the human body (*Ribeiro et al., 2012*). This database can then be used to characterize whole metagenomic data. Nevertheless, this tandem approach is still severely limited by the challenges posed by

uncultured bacteria, as evidenced by the HMP's "most wanted," a list of bacterial strains detected in the human microbiome through 16S rRNA gene sequencing that have yet to be fully sequenced (*Fodor et al., 2012*).

In order to fill these gaps, various sequencing and culture-based approaches have been pursued. Recent advances in de novo assembly of short reads, have allowed for bacterial genomes to be directly extracted from complex metagenomic samples (*Nielsen et al., 2014*; *Jeraldo et al., 2016*). The main challenge often consists in distinguishing which reads belong to one species' genome over another. *Nielsen et al. (2014)* achieve this for 238 microbial genomes by identifying co-abundant gene groups across several metagenome samples from a given environment. In some cases, the number of metagenomes in a study is enough to guarantee high-quality genomes without the need to rely on co-abundance or other methods. One such case involves 9,428 metagenomes collected for a large-scale human microbiome study spanning various continents, from which a total of 4,930 species-level genomes were reconstructed (*Pasolli et al., 2019*). Seventy-seven percent of these sequenced genomes were not previously available in public repositories. Other strategies forego metagenomics altogether. *McLean et al. (2013)*, chose to harness the recent expansion of single-cell sequencing through multiple displacement amplification to assemble "mini-metagenomes" or a set of genomes obtained from cells amplified from small random pools from a large environmental sample. This allows researchers to capture scarcer yet potentially crucial microbial species. Taking a high-throughput approach to culturing previously unculturable bacteria is yet another available strategy (*Lagier et al., 2012*). This approach, microbial culturomics, consists in the cultivation of a large number of bacterial samples originating from a single microbiome under a wide-ranging set of conditions, using mass spectrometry and 16S ribosomal RNA sequencing to rapidly identify the contents of each sample.

The combined contribution of these (*Lagier et al., 2015*, *2016*) approaches towards creating a complete set of reference genomes has increased the accuracy of taxonomic profiling, providing a springboard for further in-depth human microbiome research, particularly in the elucidation of human disease. While the HMP initially set out to uncover the composition and range of a normal human microbiome, other projects such as metaHIT expanded their investigation to include patients with conditions associated with the human intestinal tract, such as obesity and irritable bowel syndrome (*The Human Microbiome Project, 2012*; *Ehrlich & MetaHIT Consortium, 2011*). In the first ever metagenome-wide association study (MWAS), *Qin et al. (2012)* sought to identify changes in the human gut associated with type 2 diabetes by comparing fecal samples from 71 Chinese case patients against 74 controls. Since then, MWAS have been used to uncover associations between the human microbiome and a variety of other conditions including obesity, atherosclerosis, liver cirrhosis, colorectal cancer or rheumatoid arthritis (*Wang & Jia, 2016*).

Creating complete genomic reference libraries only addresses one aspect of taxonomic profiling. This growing knowledge base has to be coupled with effective strategies for exploiting it. In its earliest incarnation, taxonomic profiling consisted of attempting to align all metagenomic reads to an entire database of genome reads, specifically GenBank (*Benson et al., 2013*), using the dedicated local aligner, BLAST (*Altschul et al., 1990*).

One example of this approach being used was the pioneering sequencing study of the Sargasso Sea (*Venter et al., 2004*). However, given the necessity to align the contents of large metagenomics samples to ever-expanding microbial databases, this original strategy has become computationally intractable. Several programs have sought to address this bottleneck head on using various matching heuristics. For example, DIAMOND uses spaced seed and double indexing to provide markedly faster alignment capacity (*Buchfink, Xie & Huson, 2014*). Though these types of speedups play a large role in improving taxonomic profiling, this is by no means the only strategy to have been deployed. This challenge has led to the development in intervening years of a whole host of taxonomic classifiers based on various distinct heuristic approaches (*Breitwieser, Lu & Salzberg, 2019*).

One early solution consisted in reducing a sequence alignment to a representative subset of clade-specific marker genes, in other words, genes that could be used to reliably differentiate one species from another. For any given clade, the associated markers correspond to core genes that were derived from available reference genomes and further determined not to be present in any other outward clade (*Segata et al., 2012*). Notably, the HMP relied on such marker genes to rapidly profile species present in their metagenomes using the taxonomic classifier MetaPhlAn (*The Human Microbiome Project, 2012*; *Segata et al., 2012*). For underrepresented clades in genome databases, later iterations of MetaPhlAn, introduced quasi-marker genes. A limited number of other clades may include them in addition to its target clade. Nevertheless, they provide more accurate taxonomic profiling under the scenario in which the other clades that share those genes are determined to be absent from the metagenomic sample based on their own unique marker genes (*Truong et al., 2015*). However, these approaches are predicated on reads for these genes being successfully sampled from a microbiome, which cannot be guaranteed for species present in low abundance.

Therefore, other taxonomic profilers have been developed around the idea of using whole-genome libraries for more accurate taxonomic and functional profiling. For a program like MetaMix, this takes on the form of a preliminary assembly step using Velvet (*Zerbino & Birney, 2008*) to build contigs so that a smaller subset of sequences, whether large contigs or unassembled reads, are aligned with BLASTx (*Morfopoulou & Plagnol, 2015*). A number of other approaches involve using existing fast and low memory cost sequence alignment methods, such as those integrated in Bowtie2 (*Langmead & Salzberg, 2012*), to align reads to complete genomes. This is, for example, the case with tools like PathoScope 2.0 and Sigma (*Hong et al., 2014*; *Ahn, Chai & Pan, 2015*) that build on these alignments to estimate relative abundance. In order to further speedup reclassification, tools like Sigma carry out each alignment in parallel (*Ahn, Chai & Pan, 2015*). Whichever alignment or reference database is used, many reads end up matching several reference genomes, requiring a read reassignment step before classification occurs. Sigma performs a maximum likelihood estimation (MLE) of the relative abundance of available genomes to disambiguate reads that align to more than one reference genome (*Ahn, Chai & Pan, 2015*). PathoScope 2.0 and MetaMix, both use a Bayesian mixture-model to reclassify reads once initial alignment completed (*Hong et al., 2014*; *Morfopoulou & Plagnol, 2015*). In the case of PathosScope 2.0, reads are subjected to a mixture model

that penalizes non-uniquely aligned reads in the presence of other uniquely aligned reads (*Francis et al., 2013*). Model parameters are estimated using an expectation-maximization (EM) algorithm. On the other hand, MetaMix's mixture model is built on a parallel Markov Chain Monte Carlo approach (*Morfopoulou & Plagnol, 2015*). Taxator-tk, which like MetaMix still aligns sequences to BLAST, has overlapping elements of the search results combined into long segments which retain taxon ID information (*Dröge, Gregor & McHardy, 2015*). Each segment from a given read gets assigned a taxon identifier that corresponds to the most recent common ancestor (MRCA) deduced from the group of taxa. A final step identifies the consensus taxon to which the majority of segments in a read have been assigned.

Still other taxonomic classifiers have gone one step further by applying new compression or transformation techniques to both reference genomes and metagenomic read data to provide both fast and accurate taxonomic profiles. One popular approach consists of a prior kmerization of reference sequences, an approach implemented in Kraken (*Wood & Salzberg, 2014*). From a set of available genomes, Kraken generates a corresponding series of 31-mers, where k-mers present across multiple sequences are identified by their MRCA. By working from k-mers rather than the genome sequences themselves, Kraken significantly reduces its search space. Classification is obtained by scoring a taxonomy tree based on the k-mers that align to a read. Another pipeline, Centrifuge (*Kim et al., 2016*), also applies transformation to reference sequences via a two-step approach which results in a condensed database of references and faster access during alignment. It achieves the former by compressing highly similar strains into composite genomes, where sequences with ≥99% similarity are merged into one while divergent sequences are sequentially added to that genome representation. The resulting composite genomes are then indexed using the Ferragina-Manzini (FM) index based on the Burrows-Wheeler Transform (BWT) (*Kim et al., 2016*).

Given the variety of pipelines currently available for taxonomic classification and the various strategies they employ, a comparative analysis is needed to identify relative strengths and weaknesses of these different approaches. In this study, we compare eight pipelines that represent different strategies for taxonomic profiling, that is, marker-based, k-mer search, and read reassignment. With this analysis, we aim to determine which factors or combinations thereof most affect different taxonomic profiling strategies, assuming the most favorable conditions, that is, perfect base calling. This is coupled with an analysis of the taxonomic profilers' performance in terms of other metrics such as time and robustness faced with species absent from the reference database. Given that many new tools are published every year, and with more established tools being continuously upgraded, this study is only able to evaluate a subset of currently available tools. For this reason, all of our datasets and results are made readily available through the aforementioned Shiny app for future benchmarking with addition tools or upgrades.

## MATERIALS AND METHODS

The generation of simulated microbiome data and the evaluation of pipelines for human microbiome analyses were performed on Colonial One, a high-performance computer

cluster running Linux CentOS with a total compute capacity of 2,924 CPU cores and 1132,288 CUDA cores housed at the George Washington University. The compute nodes consist of 143 CPU nodes, including 64 dual Intel Xeon E5-2670 2.6 GHz 8-core processors and 79 Xeon E5-2650 2.6 GHz 8-core processors with a range of RAM capacity from 64 GB to 256 GB, and 32 GPU nodes with dual Intel Xeon E-2620 2.0 GHz 6-core processors with 128GB of RAM each.

## Software pipeline selection

We evaluated the performance of the following eight software pipelines in producing accurate taxonomic profiles over simulated human microbiome data:

- MetaPhlAn2 (v.2.2.0)—marker-gene (*Truong et al., 2015*)
- ConStrains (v.0.1.0)—marker-gene + variant calling (*Luo et al., 2015*)
- Sigma (v.1.0.1)—read reassignment/MLE + (variant calling) (*Ahn, Chai & Pan, 2015*)
- Taxator-tk (v.1.3.3)—read reassignment/MRCA (*Dröge, Gregor & McHardy, 2015*)
- Pathoscope 2.0 (v.2.0)—read reassignment/Bayesian (*Hong et al., 2014*)
- MetaMix (v.0.2)—read reassignment/Bayesian (*Morfopoulou & Plagnol, 2015*)
- Kraken (v.0.10.5)—k-mer search/MRCA (*Wood & Salzberg, 2014*)
- Centrifuge (v.1.0.3) —composite genome alignment (*Kim et al., 2016*)

We generated taxonomic profiles with each of these pipelines using the same reference set of complete genomes; 426 in total, from the human oral microbiome database (*Chen et al., 2010*). This microbiome was chosen for its microbial diversity, second only to the gut microbiome, with bacterial species belonging to 12 distinct phyla: Actinobacteria, Bacteroidetes, Chlamydiae, Chloroflexi, Firmicutes, Fusobacteria, Gracilibacteria, Proteobacteria, Saccharibacteria, Spirochaetes, SR1, Synergistetes (*Deo & Deshmukh, 2019*). The only exceptions were marker gene-based methods for which we used the existing database of gene marker, comparing it to our genome library of choice to ensure that all genomes used in our study were adequately represented by markers. Several of these pipelines necessitate a read alignment step. Unless stated otherwise, this particular step was performed with the aligner Bowtie2 (v2.2.9) (*Langmead & Salzberg, 2012*). Although other alignment options, such as BWA, are also available for the alignment of bacterial sequences, we chose to use Bowtie2 as it is de facto the preferred method used by various profilers in this study (*Hong et al., 2014*; *Ahn, Chai & Pan, 2015*; *Truong et al., 2015*). Based on previous research on the read alignment of bacterial genomes by several popular aligners, we do not anticipate the choice of read aligner to constitute a large confounding factor (*Thankaswamy-Kosalai, Sen & Nookaew, 2017*). Otherwise, this list of software pipelines, while non-exhaustive, runs the gamut of methods currently in use to assign taxonomic profiles to metagenomes.

MetaPhlAn2 (*Truong et al., 2015*; *Segata et al., 2012*) is a popular tool for marker gene-based taxonomic profiling. Clade-specific marker genes are used to rapidly assess the presence and abundance of taxa in complex metagenomes. For this analysis we used the gene marker database provided specifically for MetaPhlAn. The current iteration

extends the original phylum to species-level profiling to include strain-level resolution, incorporating sub-species level markers (*Truong et al., 2015*). To complete this category of taxonomic profilers, we also look at ConStrains (*Luo et al., 2015*). The ConStrains pipeline uses MetaPhlAn's original species-level taxonomic profiling as the basis for its own strain-level profiling, and unlike MetaPhlAn2, does not rely on available conspecific markers, instead performing de novo variant calling on previously aligned set of reads to tease out intra-specific diversity. Co-occurring SNVs, uncovered using SAMtools (v1.3) (*Li et al., 2009*) are combined into profiles to determine strains.

Software pipelines Sigma (*Ahn, Chai & Pan, 2015*), Pathoscope 2.0 (*Hong et al., 2014*), MetaMix (*Morfopoulou & Plagnol, 2015*), and Taxator-tk (*Dröge, Gregor & McHardy, 2015*) directly align metagenomic reads to available reference genomes, posteriorly determining to which genome each non-uniquely aligned read should be reassigned. In this category, we selected both taxonomic profilers developed around BLAST and others that take advantage of fast sequence aligners. Sigma aligns metagenomic reads to multiple reference genomes by performing each alignment in parallel, followed by an MLE of relative abundance to disambiguate reads shared by more than one genome (*Ahn, Chai & Pan, 2015*). While this MLE delivers confidence intervals for the abundance of each identified strain, we did not factor this output into our analysis given that the use of such confidence intervals is not yet common practice among taxonomic profilers. We also note that, like ConStrains, the Sigma pipeline includes a variant calling step with SAMtools to identify novel strains, but this step intervenes as part of downstream analysis and therefore was likewise not accounted for in our analysis.

Both Pathoscope 2.0 (*Hong et al., 2014*) and MetaMix (*Morfopoulou & Plagnol, 2015*) address read reassignment using Bayesian statistical modeling, the latter performing a BLASTx alignment after prior assembly using Velvet (*Zerbino & Birney, 2008*). Taxator-tk (*Dröge, Gregor & McHardy, 2015*) performs alignments using BLAST+ leaving the user the option of which BLAST algorithm to use. For this analysis, we have selected the default option, blastn. Taxator-tk then completes relative abundance estimation using MRCA.

To complete our analysis, we also included taxonomic profiling tools that use various compression techniques on reference genomes to provide fast taxonomic profiling. These approaches integrate their own read alignment step. Among the more popular profilers in that category are Kraken and Centrifuge. Kraken (*Wood & Salzberg, 2014*) generates k-mers from the reference genomes, 31-mers by default, and identifies each of these segments by its MRCA. This provides an additional opportunity to compare the performance of two taxonomic profilers that use MRCA with different alignment strategies, the other being Taxator-tk. Centrifuge (*Kim et al., 2016*) is also self-contained, compressing highly similar strains into composite genomes and aligning them using the BWT found in several aligners. For both these strategies, default parameters were largely followed.

Each of the pipelines, with the exception of MetaMix, was run on 16 threads using the default parameters as set by the respective developers. For MetaMix, we applied 16 threads to the BLASTx search step, but could not apply more than 12 cores to the rest of the

pipeline due to software limitations. MetaMix crashed when given datasets that simulated sequence depths of 10 million or more reads.

## Generation of simulated data

In order to evaluate and compare the efficacy of different pipelines to correctly assign taxonomic profiles under a number of different scenarios, we simulated sequence reads from a metagenome using MetaSim (*Richter et al., 2011*). In this evaluation, we are assuming error-free sequencing, with no a priori information about the platform used. In other words, we are testing each pipeline under ideal base calling conditions to then be able to focus on attributes of the metagenome and the underlying microbiome itself, where the former is a fairly accurate snapshot of the latter. While introducing sequencing error would certainly affect performance, with methods that integrate quality score data faring better than others, the attributes of the metagenome already provide ample parameters on which taxonomic profiling methods may differ. We centered our analysis on read properties such as read length and sequencing depth, as well as features of the metagenome such as species dominance and number of species (described in more detail below). The taxonomic profilers selected for this study were developed to handle data from sequencing technologies with relatively short reads, we have therefore limited our analysis to reads between 75–1,000 bp. The simulated datasets are based on the 426 complete or permanent draft genomes obtained present in the Human Oral Microbiome Database at the time the analysis was first conducted (*Chen et al., 2010*). As a result, we generated a total of 144 simulated metagenome datasets representing every possible combination of attributes. How their parameters vary across datasets is summarized in Table 1.

In terms of metagenome attributes, the two main parameters considered were number of species and dominance. The highest possible value for distinct species number reflects an upper limit imposed by the availability of complete genomes in the database. The four dominance scenarios mimic common scenarios encountered when dealing with human microbial data. A single species present in half of all reads typically suggests acute infection. Conversely, the presence of species in roughly equal number is our null hypothesis. Our null hypothesis corresponds to the proportions used in controls, a convention followed by groups such as ATCC (https://www.atcc.org/Microbiome) and Zymo (https://zymoresearch.com/pages/m-sci). We also study dominance scenarios where a small group of species show high abundance (10% → 25%) as can be observed during a polymicrobial infection, as well as one where half of the species account for 80% of the reads (50% → 80%). This last scenario simulates a regular non-dysbiotic microbiota. In total, 12 scenarios for metagenomic compositions are tested.

For each of these hypothetical metagenomes, we simulated reads of different sizes, reflecting differences in read size among different technologies and sequencing depths. The combination of all possible metagenome and sequencing parameters lead to a testing space of 144 datasets on eight different pipelines or 1,152 discrete scenarios. Due to computational limitations inherent to the R package, we were not able to test MetaMix for our highest sequencing depth; 10M, reducing the number of discrete results down to 1,104. With

**Table 1 The attributes around which the simulated datasets were generated.**

| Dataset attributes | Parameters | Total for attribute |
|---|---|---|
| Metagenome/microbiome attributes | | |
| Species present in metagenome | 10; 100; 426 | 3 |
| Species dominance scenarios | 1→50%; 10→25%; 50→80%; all equal | 4 |
| Sequencing attributes | | |
| Read length (bp) | 75; 150; 300; 1,000 | 4 |
| Sequencing depth (reads) | 100K; 1M; 10M | 3 |
| Total number of datasets | | 144 |
| Software Pipelines | | |
| MetaPhlAn2; Constrains; Sigma; Taxator-tk; PathoScope 2.0; Metamix*; Kraken; Centrifuge | | 8* |
| Total number of taxonomic profiles in analysis | | 1,104 |

Notes:
Dominance scenarios are represented in the following form (number or percentage (%) of species belonging to the most represented group –> cumulative abundance in dataset).
* Datasets with an average sequencing depth of 10M could *not* be generated for MetaMix

these datasets (available at: https://github.com/microgenomics/HumanMicrobiomeAnalysis), we carried out a multipart analysis, detailed below. With some analyses, additional simulated datasets were necessary. These will be specified in the appropriate subsections below.

## Performance evaluation

We assessed the sensitivity and specificity of the taxonomic profiles of the selected pipelines under the 144 different scenarios simulated with our metagenomics datasets. Sensitivity was calculated using the proximity of each estimated abundance for a given genome to its true simulated abundance. We considered the estimated abundance to be a match for true simulated abundance if that measure fell within ±50% of the actual value for a given species. It is inherently difficult for a taxonomic profiler to perfectly predict actual abundance. We therefore include a range within which an estimate can be considered correct enough for practical purposes. Specificity was calculated based on the absence of abundance estimates for species absent from the simulated data.

In addition to measuring sensitivity and specificity, we also assessed abundance estimates produced by different pipelines against actual abundance using relative root mean squared error (RRMSE) and average relative error (AVGRE), given by Eqs. (1) and (2):

$$\text{RRMSE} = \sqrt{\frac{1}{K}\sum_{j=1}^{K}\left(\frac{|w_j - t_j|^2}{t_j}\right)} \tag{1}$$

$$\text{AVGRE} = \frac{1}{K}\sum_{j=1}^{K}\left(\frac{|w_j - t_j|}{t_j}\right) \tag{2}$$

where, for $K$ genomes, $t_j$ and $w_j$ are, respectively, the true and estimated abundance of species $j$.

## Additional measures simulating further datasets

For our final two performance metrics, we focused our testing on five of the eight original software pipelines presented here, measuring the effect of genome absence from the relevant database on taxonomic classification, as well as computational run time.

We excluded ConStrains on the basis that results were bound to be too similar to those of MetaPhlAn2, the latter therefore acting as a useful proxy for the former. We also excluded MetaMix given execution problems passed a certain sequencing depth. Finally, we excluded Sigma given its low sensitivity. For each of these metrics, additional simulated datasets were generated. We detail those in the following sections.

## Genome absence abundance effect

Up to this point in our analysis, we have provided taxonomic profilers with a complete reference library to work from, providing either a marker gene or genome reference for each of the species with a possibility of being present in the simulated metagenomes. However, this approach does not account for the possibility of unknown species being present in a metagenome. Consequently, we do not have a measure of the effect such a species would have on each pipeline's abundance estimates. In order to measure the potential effect reads from a genome absent from our reference database might have on taxonomic profiling, particularly pertaining to known bacterial strains, we generated new datasets that included reads from the bacteria *Bacillus cereus*. Similar strains already exist in the database, including *Bacillus anthracis*, *Bacillus daussi*, and *Bacillus subtilis*.

We therefore compared known simulated abundance values for these three taxa with estimated values before and after the addition of reads from the *Bacillus cereus* genome.

## Computational run time

To determine the efficiency (computational run time) of each pipeline and how it scales with the addition of more reads, we measured computational time in CPU minutes for a range of different sequencing depths. For this, we took two existing datasets which both include 426 species, reads of 150 base pair length and where a few species show high abundance ($10\% \rightarrow 25\%$), with sequencing depths of 1M and 10M, respectively. We then generated 13 new datasets, sharing those same attributes with the exception of sequencing depth which were multiples of $10^6$ and $10^7$, in the latter case up to $6 \times 10^7$. The times were given in real compute time rather than clock time to account for the time complexity necessary to run taxonomic profiles.

## RESULTS

Initially, eight software pipelines were assessed using 144 simulated metagenomics datasets producing a total of 1,104 taxonomic profiles from which sensitivity and specificity metrics were calculated. MetaMix was unable to handle the high volumes of data associated with datasets with a simulated depth of sequencing of 10M reads and therefore no metrics were derived for datasets of this sequencing depth or higher. Given the range and high-dimensionality of the data generated for this analysis, we have made our results

available in the form of dynamic graphs using an R Shiny application (https://github.com/microgenomics/HumanMicrobiomeAnalysis). The application allows the reader to control which data points are displayed based on the dataset and pipeline from which the data were generated. The data are also available for download allowing researchers to directly compare new tools using these same data to the tools evaluated in this study. In this section, we provide the results for each of our tests and encourage the reader to follow those with the appropriate graphical tab.

## Taxonomic profiling: sensitivity and specificity

Each pipeline draws a distinctive sensitivity and specificity pattern across the numerous scenarios tested in our analysis. MetaPhlAn2 and ConStrains are highly correlated as the latter builds on the former, but they are the exception to the rule. The bulk of variation between pipelines revolves around sensitivity. The values recorded in this analysis cover nearly the entire range of possible values from 0.017 up to 0.974. The marker gene approach, covered by MetaPhlAn2 and ConStrains, accounts for all values <0.25, producing the 40 taxonomic profiles with the lowest sensitivity in our analysis (Fig. 1A). This already suggests some limits to what can be achieved with a marker gene approach. Excluding these two pipelines and Sigma, all other pipelines produce taxonomic profiles which reach a sensitivity of at least 0.5. For these remaining pipelines, only two profiles fall below this threshold, one for Taxator-tk and the other for PathoScope 2.0 (Figs. 1B and 1C). At the other end of the range are taxonomic profiles with sensitivity >0.95 produced by PathoScope 2.0 and MetaMix (Fig. 1C), the two taxonomic profilers that perform read reassignment using a Bayesian approach. Contained within that range are all of PathoScope 2.0's profiles based on simulated datasets that include all 426 available species, suggesting that this method is highly suited to the study of diverse metagenomes. It is worth noting that, while Centrifuge and Kraken do not produce taxonomic profiles in that range, a significant proportion of their taxonomic profiles reach a sensitivity >0.90 (Fig. 1D). In Centrifuge's case, sensitivity never drops below 0.75. Both tools show the least amount of pipeline-specific variation in sensitivity with standard deviations of 0.029 and 0.097 respectively (Fig. 1D). By contrast, MetaPhlAn2 and Constrains have the highest pipeline-specific variability, both with a standard deviation of 0.187 (Fig. 1A).

While differences in specificity certainly exist, we observe less of a spread between pipelines, ranging from 0.777 to complete specificity, than with sensitivity. Much of the lower-end variability in specificity can be attributed to single pipeline: Taxator-tk. The pipeline's taxonomic profiles for datasets containing 426 species are the least specific encountered in this analysis (mean: 0.82; Fig. 1B). To a lesser extent, Centrifuge also contributes some of the less specific profiles present in this analysis. Datasets of medium to high sequencing depth; 1–10M and containing 10 or 100 species lead to Centrifuge's lowest specificity profiles (mean: 0.859), although this interval also contains a few taxonomic profiles from MetaPhlAn2, ConStrains and MetaMix. Incidentally, Taxator-tk and Centrifuge are also the only two pipelines in this analysis that do not reach full specificity under any scenario, although it is worth noting Centrifuge does produce highly specific

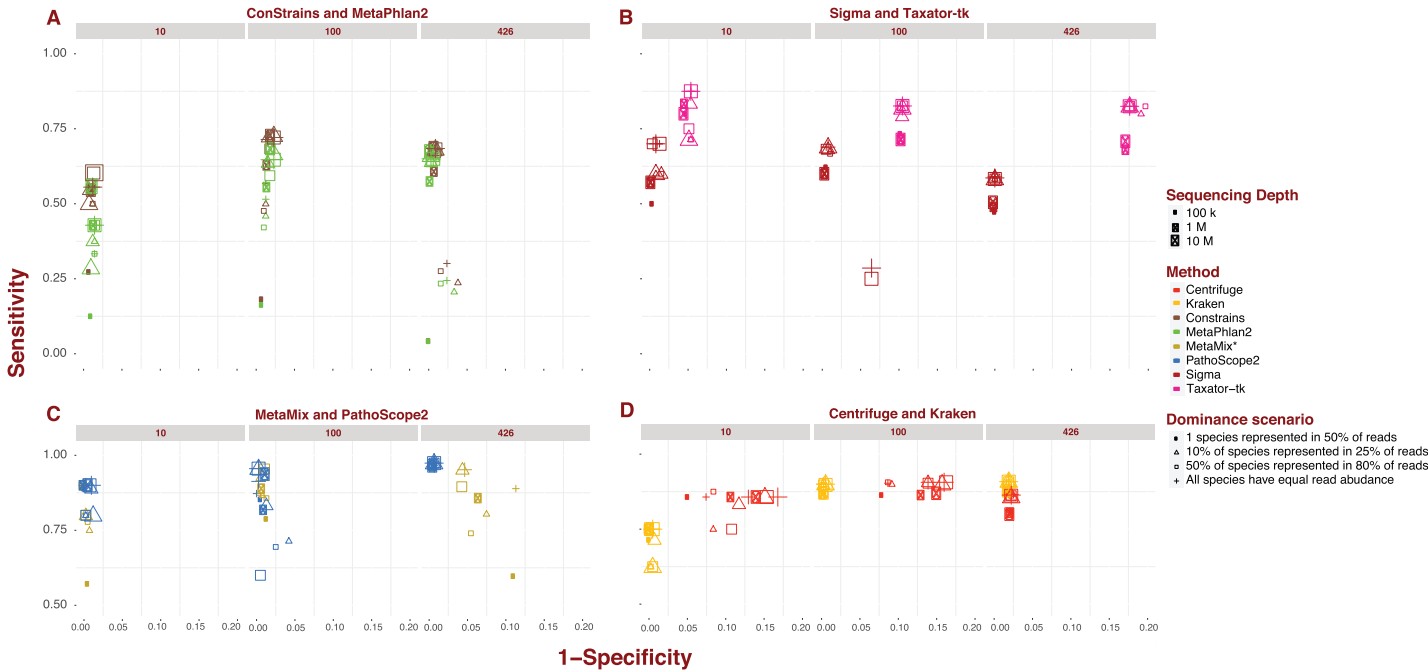

**Figure 1 Sensitivity and specificity of tested software under diverse scenarios using 150 bp reads.** (A–D) show sensitivity and specificity for the eight software implementations for three different sequencing depths (100K, 1M, and 10M reads; size of the points) and four dominance scenarios (shape of the points). (A) shows results for marker gene approaches; (B) for whole genome approaches; (C) for approaches based on read reassignment; and (D) for approaches that implement reference library transformations.

taxonomic profiles for datasets that include 426 species (mean: 0.978). In contrast, Taxator-tk reaches its peak with profiles of 0.954. On the other end, pipelines such as Kraken and Sigma, in addition to reaching full specificity under a number of scenarios, show little to no variation between profiles, with specificity dropping no lower than 0.978 (means: 0.991) and 0.93 (mean: 0.994) respectively. At the very least this suggests that most taxa identified in a profile are indeed present in the metagenome, even when their abundance is not correctly given. To get a better understanding of how sensitivity and specificity vary within and between pipelines and strategies, we will now look at the different pipelines in more depth.

## Marker gene approach: MetaPhlAn2 and ConStrains

Overall, ConStrains provides more sensitive results than MetaPhlAn2, but this conclusion is by no means applicable to every dataset in isolation. Both occupy a similar range of sensitivity values from 0.017 up to 0.684 and 0.727 respectively and, together with Sigma at 0.7, have the lowest upper-end values of the entire analysis (Figs. 1A and 1B). For datasets containing 100 or 426 species, the attribute with the strongest impact on MetaPhlAn2 sensitivity is sequencing depth. This is particularly striking for datasets containing all 426 species and obtained at a low sequencing depth of 100K, showing consistently lower sensitivity (mean: 0.144; 0.171) than with 1M (mean: 0.624; 0.654) or 10M (mean: 0.661; 0.683). In each case, ConStrains provides some marginal improvements in sensitivity. With 100 species, sensitivity is still noticeably lower at sequencing depth of 100K

(mean: 0.359; 0.409) than at 1M (mean: 0.612; 0.687) or 10M (mean: 0.666; 0.725), but with some small amount of overlap.

The presence of many species in low abundance in a metagenome, has a similar negative impact on sensitivity as sequencing depth. By way of comparison, taxonomic profiles from datasets of low sequencing depth that uniformly represent species, our null case, overlap in sensitivity with profiles from MetaPhlAn2 of medium sequencing depth for which a single species dominates half of the reads, with other species represented in low abundance. Generally, at low and medium sequencing depth, datasets where a single species is present in half of the reads lead to lower sensitivity; all else being held equal. This is again most striking when the number of species is high; 426, and sequencing depth low (mean: 0.029; 0.03), but it applies to datasets with 100 species as well (mean: 0.125; 0.144). What we are observing here again is a lack of adequate read representation for most species in low abundance. Low sequencing depth also leads to increased susceptibility to read length, shorter reads leading to better sensitivity than longer reads; all other attributes being held equal.

With datasets containing 10 species, the previous observations do not apply as cleanly, likely due to the larger share of reads per species. The first noticeable difference is that there is less variation in sensitivity between datasets ranging from 0.125 to 0.556 for MetaPhlAn2and from 0.2 to 0.636 for ConStrains. The lower end of this sensitivity range is largely imputable to taxonomic profiles of low sequencing depth; 100K, where a single species dominates reads (mean: 0.198; 0.236). Beyond that it is difficult to draw many conclusions given the irregularity with which the data appear to vary in sensitivity, with dominance scenarios being more closely related when there are only 10 species in a metagenome. In terms of specificity, both pipelines vary across a smaller range from 0.883 and 0.885 up to full specificity, remaining largely correlated. ConStrain's specificity is overall better or identical to MetaPhlAn2's for datasets containing 10 or 100 species and worse for those containing 426, although often by a negligible amount. For both pipelines, the taxonomic profiles with the highest and lowest specificity by and large correspond to datasets that contain all 426 species. For such datasets, the lowest specificity involves the subset of profiles with both low sequencing depth; 100K and long reads; 1,000 bp (mean: 0.902 for both) and 300 bp reads (mean: 0.939; 0.94), although not including datasets where a single species is present in half of the reads. For the latter dataset, we observe high specificity (mean: 0.996 for both), most likely as a function of the absence of reads corresponding to marker genes for species of low abundance. It is harder to assign particular attributes to high specificity, although the pipelines reach high specificity for datasets with the highest sequencing depth; 10M, with specificity consistently higher for MetaPhlAn2 (mean: 0.996; 0.991). With a few exceptions, we also observe high specificity for medium sequencing depth (mean: 0.991; 0.987). By comparison, specificity varies much less for datasets containing either 10 species (mean: 0.987; 0.989) or 100 species (mean: 0.982 for both). Most of the variation in results for MetaPhlAn2 and ConStrains is therefore centered on sensitivity, although the pipelines are in general both more sensitive and specific for higher sequencing depths.

## Starting from whole genome libraries: Sigma and Taxator-tk

Provided we exclude the four outlying taxonomic profiles among datasets containing 100 species, Sigma's sensitivity, ranging between 0.429 and 0.7, appears to be among the least variable, while its specificity is by far the least variable going from 0.986 to full specificity (Fig. 1B). However, Sigma's sensitivity is also the lowest for the class of tools involving read reassignment. Overall, datasets that include 100 species lead Sigma to more sensitive taxonomic profiles (mean: 0.654), again excluding outliers, than datasets that include 426 species (mean: 0.56). In both cases, we see a noticeable difference in sensitivity when a single species is present in half of the reads; (mean: 0.593) for 100 and (mean: 0.487) for 426, than under other dominance scenarios; (mean: 0.676) for 100 and (mean: 0.584) for 426. There is otherwise little to no variation in sensitivity or specificity between taxonomic profiles from datasets that include 100 or 426 species; excluding the aforementioned outliers.

With the exception of the four outliers, specificity is always high (mean: 0.996), profiles reaching full specificity for datasets that include 426 species. What diversity Sigma does capture, is therefore largely correct, even if the abundance of some species is incorrectly estimated.

Taxator-tk produces taxonomic profiles over a wide range of sensitivity values, from 0.333 up to 0.875, although it is important to note that only one taxonomic profile falls below 0.5 (Fig. 1B). The lower end of that range is largely due to datasets composed of short 75 bp reads, leading to lower sensitivity (mean: 0.681) than the bulk of datasets with longer reads (mean: 0.816). All other attributes being equal, a dataset with 75 bp reads will systematically lead to a less sensitive taxonomic profile when using Taxator-tk. The difference is particularly pronounced for datasets containing only 10 species. Among datasets that contain 100 or 426 species we also see consistently lower sensitivity for datasets where a single species is present in half of reads (mean: 0.736) than under any other dominance scenario (mean: 0.839). For the latter datasets, the positive effect of longer read size becomes more apparent; (mean: 0.757) for 75 bp reads, (mean: 0.821) for 150 bp reads, (mean: 0.838) for 300 bp and (mean: 0.857) for 1,000 bp reads. Unlike the previous approaches, Taxator-tk is built around BLAST. Longer reads are therefore better suited to the task. As with Sigma, Taxator-tk takes advantage of the entire set of reads available in a metagenome, ensuring that relatively good results can be obtained whatever the sequencing depth and for most dominance scenarios, provided large read length.

While Taxator-tk's taxonomic profiles are fairly sensitive given the right conditions, it generally lacks in specificity falling as low as 0.778 and rising no higher than 0.954. Taxator-tk's profiles overall have lower specificity than all other taxonomic profilers, most overlap in terms of specificity occurring with Centrifuge for datasets containing 10 or 100 species. Specificity is particularly susceptible to the number of species in the dataset with higher specificity occurring with lower species diversity; (mean: 0.95) for 10 species, (mean: 0.896) for 100 species and (mean: 0.82) for 426. Overall, Taxator-tk appears to be most susceptible to read length and species diversity, the first improving sensitivity and

the other lowering specificity as they increase in size. This is likely due to the number of species present in a metagenome complicating the MRCA process. Compared with gene-marker strategies and Sigma, Taxator-tk is a markedly more sensitive taxonomic profiler, provided reads are at least 100 bp in length. However, sensitivity still has an upper bound of 0.875, setting it apart from the methods that follow. Additionally, while Taxator-tk offers more sensitivity it comes with a non-negligible cost in terms of specificity.

## Using Bayesian models: PathoScope 2.0 and MetaMix

PathoScope 2.0's taxonomic profiles for the most part occupy the upper-end of the range of sensitivity values observed in this analysis, between 0.467 and 0.974, with only a single profile showing sensitivity <0.5 (Fig. 1C). The lower range of values corresponds to datasets that include 100 species (from 0.467 to 0.956, mean: 0.811). It is also within this range that we see the most variability, particularly for sequencing depths of 100K or 1M (mean: 0.749). At these sequencing depths, 1,000 bp reads lead to less sensitive results (mean: 0.617) compared to shorter reads (mean: 0.793). We also see lower sensitivity when half of the species in a simulated metagenome account for 80% of reads (mean: 0.64). For comparison, PathoScope 2.0 produces more sensitive results when species are represented equally in reads, the null case, and reads are shorter than 1,000 bp (mean: 0.892). In contrast to all the variability observed at 100K and 1M depth, high sensitivity is guaranteed with a high sequencing depth of 10M reads (mean: 0.936), with only a single profile below <0.9. PathoScope 2.0 reaches both its highest sensitivity and lowest variability with taxonomic profiles derived from datasets containing all 426 available species (mean: 0.971), a range that no other pipeline occupies in this analysis. For datasets that include 10 species, PathoScope 2.0 displays a small range of sensitivity (mean: 0.866) more closely resembling sensitivity for datasets with 426 species. In general, PathoScope 2.0 produces some of the most sensitive taxonomic profiles recorded in this analysis. Crucially, PathoScope 2.0 produces its best results on the most diverse profiles. Within that configuration, its Bayesian approach produces highly accurate relative abundance estimates, whatever the metagenome composition, sequencing depth or read length.

Specificity for PathoScope 2.0 is confined to a range between 0.9 and 0.994. This also corresponds to the range observed for datasets including 426 species. It should be noted however, that most of the variability within that group is imputable to datasets combining 1,000 bp reads and low sequencing depth of 100K (mean: 0.949), remaining datasets displaying a sensitivity of 0.994. As with sensitivity, specificity for datasets that include 100 species is more contrasted with values ranging from 0.919 to full specificity. A large fraction of the lower end specificity corresponds to datasets with 1,000 bp reads (mean: 0.942) compared to (mean: 0.99). Finally, there is much less variability in specificity for datasets containing 10 species (mean: 0.995). In this case, the attribute that has the biggest impact is sequencing depth, with 10M leading to lower specificity (mean: 0.991 compared to 0.997). It is interesting to observe a dip in overall sensitivity for metagenomes with 100 species, compared to both low and high diversity metagenomes. However, it is worth pointing out that even with 100 species, most profiles remain highly sensitive and specific.

We can only speculate that the internal logic of PathoScope 2.0's read reassignment Bayesian algorithm makes it less sensitive when 80% species are represented by 50% of reads when confronted with 100 species.

Like PathoScope 2.0, MetaMix taxonomic profiles can reach a high-level sensitivity under specific conditions (Fig. 1C). However, also like PathoScope 2.0, sensitivity varies across a wide range of values from 0.557 up to 0.956. In general, datasets of low sequencing depth; 100K, lead to less sensitive taxonomic profiles (mean: 0.786) than at medium depth; 1M (mean: 0.894), provided species are not represented in equal abundance. In that particular case, sensitivity varies much less between 100K (mean: 0.886) and 1M (mean: 0.933). The combination of 100K sequencing depth and one species being represented in half of reads leads to particularly low sensitivity for datasets of 10 and 426 species (mean: 0.6). All other datasets occupy a higher range. We were unable to determine sensitivity for a higher sequencing depth of 10M because of pipeline limitations addressed in the methods section. MetaMix relies on prior de novo assembly using Velvet before aligning sequences with BLAST. This both explains why MetaMix does not show the same variability over read length than Taxator-tk and why the taxonomic profiler performs less well with low sequencing depth, which would lead to less successful assembly, with the non-assembled reads aligned directly to BLAST in greater numbers. As with Taxator-tk, MetaMix specificity is largely a function of the number of species included in the dataset with datasets that include 10 species more specific (mean: 0.996) than those that include 100 species (mean: 0.989), all of which are more specific than datasets that include 426 species (mean: 0.926). The range of possible values also increases with the number of species. However, unlike Taxator-tk, specificity remains relatively high even when species count is high. With both PathoScope 2.0 and MetaMix, we see a marked increase in sensitivity compared to previously described methods that does not come at a large cost in terms of specificity. Both pipelines produce profiles which by and large achieve a sensitivity of at least 0.5 and, under the right conditions, rise to above 0.9.

## Reference library transformations: kraken and centrifuge

Compared to other pipelines studied in this analysis, Kraken and Centrifuge show much less variation in sensitivity (Fig. 1D), particularly once we focus our attention on datasets containing either 100 or 426 species. With Kraken, we obtain profiles going from 0.625 up to 0.91 in sensitivity. For datasets containing 10 species, Kraken's taxonomic profiles are of medium sensitivity (mean: 0.705). However, once we consider only datasets of 100 or 426 species, sensitivity increases significantly (mean: 0.896). The lower end of this range (mean: 0.872) contains exclusively profiles produced from datasets where a single species dominates half of reads, with other dominance scenarios contained in a narrower bracket (mean: 0.904). These latter profiles can be further separated between those that originate from datasets of 100 species (mean: 0.9) and those that include 426 species (mean: 0.908), this latter cross-section representing Kraken at its most sensitive. Kraken, therefore, produces high quality profiles given a fairly diverse metagenome, whatever other attribute this metagenome takes. The use of reference sequence k-mers obviates the need for posterior read reassignment. This can be both an advantage and a limitation,

as inevitable k-mer collision cap an otherwise high sensitivity. This could explain why, while Kraken performs overall better than other taxonomic profilers, it does not always reach the sensitivity levels of other tools such as PathoScope 2.0 and MetaMix.

Kraken also varies little in terms of specificity, from 0.978 up to full specificity, which once more speaks to its robustness. Lower specificity is associated with the number of species, with datasets that include 426 species sharing a specificity of 0.981, with a single exception; 0.978. For datasets that include 100 species, the resulting specificity is high (mean: 0.996). Specificity is higher still for datasets that included no more than 10 species (mean: 0.997). Overall, Kraken's sensitivity and specificity appears most affected by number of species in the original dataset. In this analysis, Kraken appears to be the least variable pipeline, varying little across either sensitivity or specificity. This is coupled with an overall high sensitivity and specificity.

Centrifuge shows the least amount of variation in sensitivity across tested scenarios, ranging from 0.75 to 0.914. As with Kraken, the lower end of that range is imputable to datasets containing only 10 species (mean: 0.843), a higher number of species leading to both higher and less variable sensitivity between taxonomic profiles (mean: 0.869). Interestingly, peak sensitivity is reached for 100 species (mean: 0.892) rather than 426 (mean: 0.846). As with Kraken under similar conditions, Centrifuge's sensitivity is degraded by the dominance of one species in a half of all simulated reads; (mean: 0.854) for 100 species and (mean: 0.8) for 426 species, compared with other dominance scenarios which occupy the rest of these respective ranges. For the latter scenarios, short 75bp reads leads to visibly more sensitive profiles for both 100 species (mean: 0.911) and 426 (mean: 0.873) then other read lengths; (mean: 0.902) and (mean: 0.857) respectively.

Unlike most other pipelines, Centrifuge never reaches full specificity and varies across datasets, from 0.832 up to 0.978, to an extent only outdone by Taxator-tk. Going against the trend observed with many other pipelines, Centrifuge reaches its highest specificity with datasets that include all 426 species (mean: 0.978). For datasets containing 10 or 100 species, specificity varies widely (mean: 0.88). The attribute with the highest impact in this case is read depth, with a higher read depth translating to lower specificity; (mean: 0.844) for 10M, (mean: 0.874) for 1M and (mean: 0.921) for 100K. With one exception, Centrifuge's profiles are of lower specificity when datasets include 100 species than with 10 species all other things held equal. Centrifuge's unusual distribution of specificity values may be the product of its reference compression approach, which working best with the highest number of species. This is also its strength however, deriving a high sensitivity that varies little between scenarios, particularly for datasets containing over 100 species. Both Kraken and Centrifuge reach high sensitivity and succeed in remaining sensitive under several conditions. For Centrifuge this comes at some cost in terms of specificity, but it is a justifiable loss given the added sensitivity, and as we shall see further below, increased speed.

## Standard error
For each of our eight software pipelines, we measured the deviation of estimated abundance from true abundance using both average error and relative root mean square

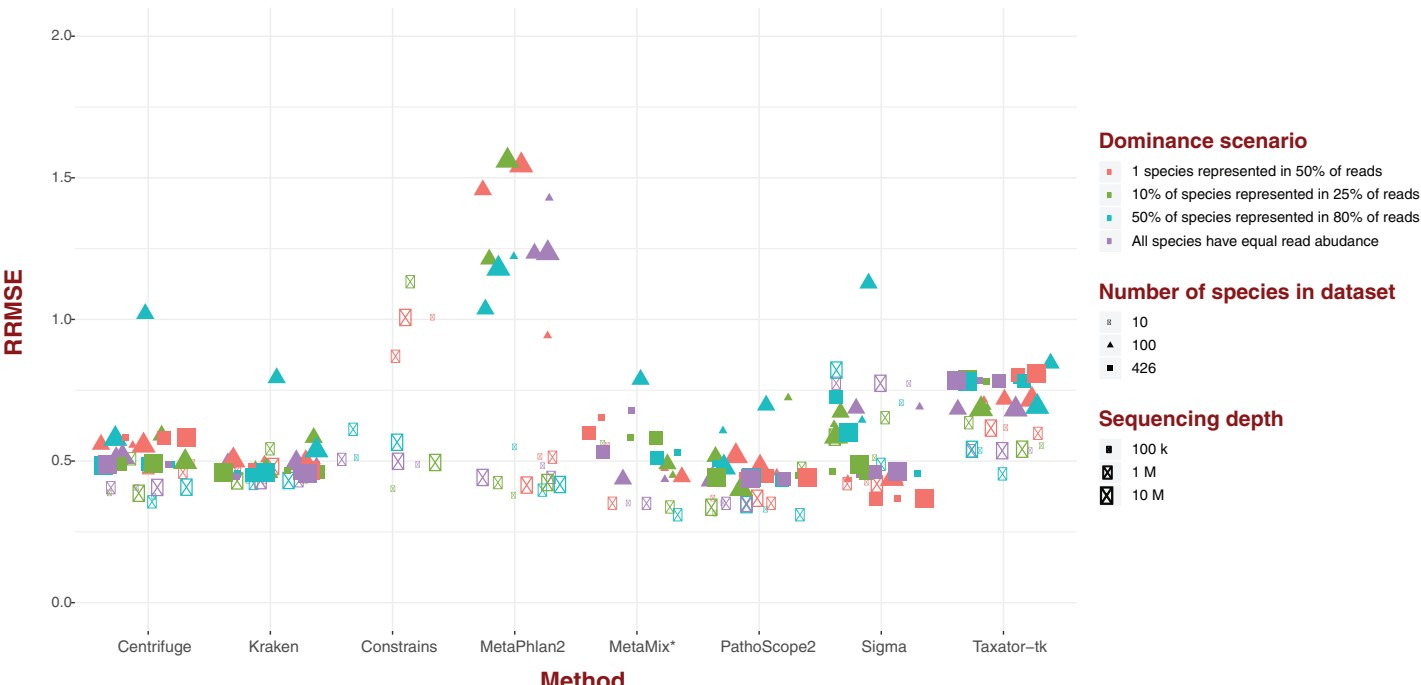

**Figure 2 Relative Root Mean Square Error (RRMSE) for every dataset of read length 150 bp grouped by software pipeline.** For each pipeline, datasets tend to group around an RRMSE of 0.5 or less, with notable exceptions: Taxator-tk, MetaPhlAn and Constrains, with the two latter cases having a number of datasets that could not be represented here. Sigma also has a few outliers.

error (RRMSE; Fig. 2). Values for RRMSE cover an extremely large range (0.301–136.644), but most of the extreme values emanate from MetaPhlAn2 and ConStrains, with all taxonomic profiles with a RRMSE >2.9 belonging to either (Fig. 2). With the exception of a few outliers in the case of Sigma, taxonomic profiles emanating from all other six pipelines display an RRMSE below 1, with nearly all taxonomic profiles gravitating around an average of 0.5. Here, the exception is Taxator-tk with an average nearer to 0.65. These results paint the same picture with regards to MetaPhlAn2, ConStrains and Sigma, the former two having the most variance compared to other taxonomic profilers, with Sigma showing lower variance that is still higher than other pipelines in the analysis.

## Genome absent from database

For the five pipelines we selected for further testing, we recreated every possible dataset with an additional "unknown" bacterial genome, *Bacillus cereus*. Crucially however, we did not incorporate a corresponding reference to our genome library, thus simulating the absence of representation of a genome present in metagenomic data. We illustrate the impact of an unknown genome for datasets containing 426 species, a sequencing depth of 10M, and read of 150 bp length under different dominance scenarios (Fig. 3). Under most scenarios, some reads for *B. cereus* are identified as reads for *B. anthracis*, thus increasing the latter's abundance estimate, whatever the original estimate. Critically, for datasets containing 426 species, and therefore containing reads from *B. daussi* and *B. subtilis* in non-negligible amounts, we still only strictly observe increases in *B. anthracis* assignments.

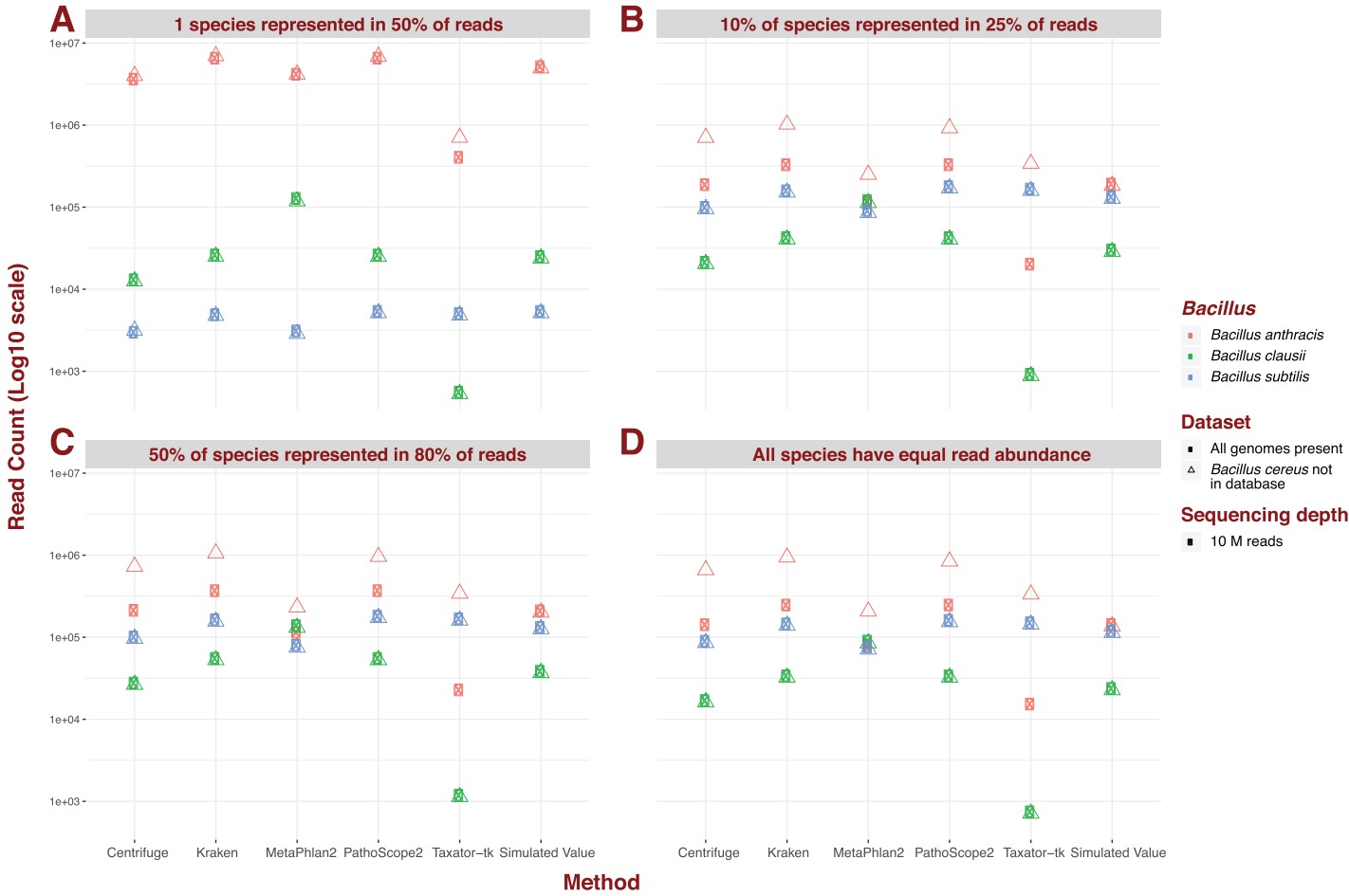

**Figure 3** **Example of the difference in *Bacillus* estimated abundance in datasets with (complete) and without (one more) the unreferenced *Bacillus cereus*.** Here, number of species (426), sequencing depth (10M) and read length (150 bp) are held constant. The largest change in abundance arises in *Bacillus anthracis*, the closest to *Bacillus cereus*. (A) One species represented in 50% of read abundance (e.g., acute infection); (B) dominance for 10% of species represented in 25% of the reads (e.g., polymicrobial infection); (C) 50% of species represented in 80% of the reads (e.g., normal microbiota); and (D) all species exhibit equal read abundance.

The presence of the unreferenced *Bacillus* species only affects the phylogenetically closest referenced species *B. anthracis* (*Bazinet, 2017*). Under a dominance model of one species being present in half of reads, some results produced by MetaPhlAn2 do not follow the trend outlined previously. Instead sensitivity of *B. anthracis* can vary wildly from not detected when *B. cereus* is present to only being detected when this extra bacterium is present.

## Real computing time

Real computing time was measured for a range of sequencing depths for five distinct pipelines (Fig. 4). The tested pipelines can be divided into two categories: pipelines that involve either gene markers or reference genomes transformation and pipelines that perform alignment on complete genomes directly. In this first category, Centrifuge performs consistently better than all the pipelines evaluated, clocking around 13 s for a sequencing depth of 1M and just under 15 min for a depth of $6 \times 10^7$, the highest depth considered in our analysis (Fig. 4). Both Kraken and MetaPhlAn2, and by extension

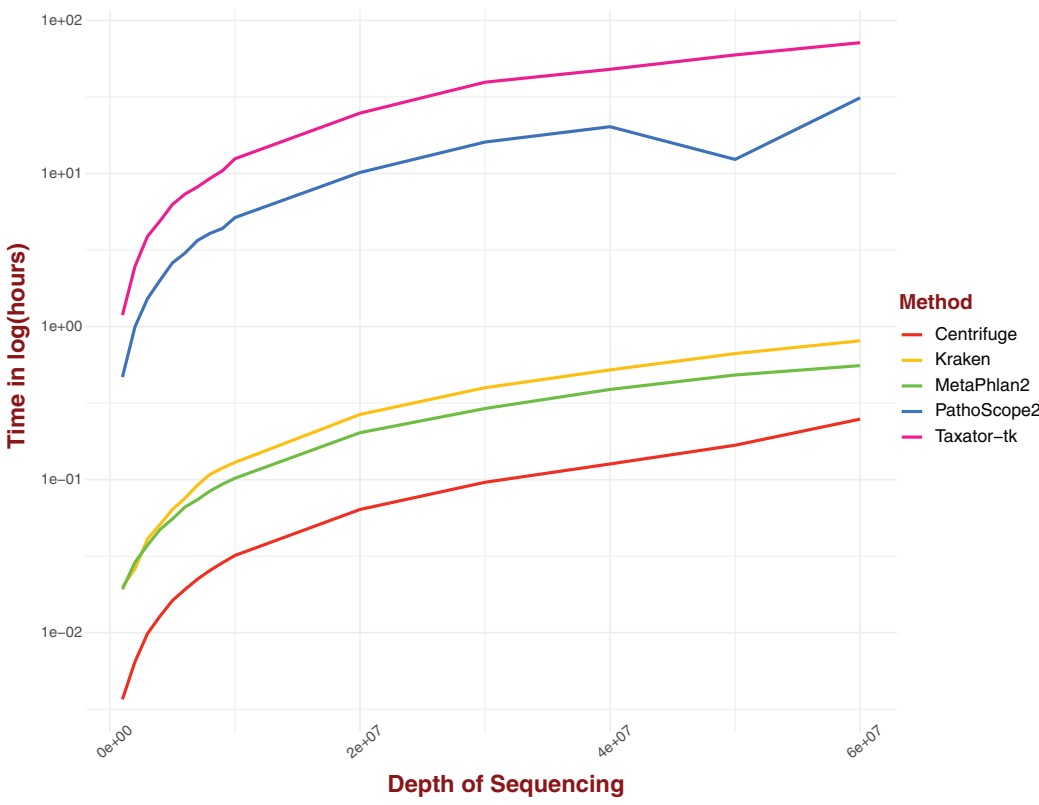

**Figure 4 Real computing time for five taxonomic profilers over 15 datasets with a range of sequencing depths with time given in ln(hours).** Pipelines that use marker gene or otherwise transform reads/references into new structures fare better than those methods that use complete genomes without alteration.

ConStrains, perform well, with both algorithms displaying similar real computing time at lower sequencing depths; 72 and 69 s respectively for a sequencing depth of 1M (Fig. 4). However, MetaPhlAn2's real computing time increased more slowly with rising sequencing depth, performing better than Kraken by almost 15 min at a maximum sequencing depth of $6 \times 10^7$; 33 and 48 min respectively. By contrast, pipelines that involve alignment against an entire set of complete genomes without any library selection or transformation, such as PathoScope 2.0 or Taxator-tk typically require much more computing time, particularly the latter. At a sequencing depth of 1M, PathoScope 2.0 requires 28 min to run, while Taxator-tk requires over 71 min. At the maximum test sequencing depth of $6 \times 10^7$, real computing time is a little over 31 h and a little over 71½ h respectively. In all cases, the rate of real computing time increases rapidly before slowly leveling off, most likely due to the limit in possible reads for all species present in the datasets. What MetaPhlAn2 loses in accuracy it makes up in running time, performing much better than programs such as PathoScope2 and Taxator-tk with their overall higher sensitivity. Here, we observe a necessary trade-off between time and sensitivity. However, both Kraken and Centrifuge succeed in providing some of the most sensitive profiles while simultaneously holding a running time that challenges that of MetaPhlAn2 in the case of the former and surpasses it in the case of the latter.

## DISCUSSION

The generation of 1,104 taxonomic profiles for the assessment of eight software pipelines has allowed us to establish which metagenome characteristics most affect specific methods, and which broadly have little effect. Measures of sensitivity and real computing time were particularly informative in separating the various taxonomic profilers in distinct categories.

An important factor in sensitivity variation, both for and between taxonomic profilers, is the availability of reads for each species present. This factor manifests itself through sequencing depth, dominance scenario, and metagenomic diversity. As a general rule, taxonomic profilers produce less sensitive profiles for dominance scenarios corresponding to acute infection in metagenomes containing 100 or more species, all else being equal. This is most likely the direct consequence of there not being enough reads to accurately characterize abundance for the overwhelming majority of species present in low abundance. Otherwise, not all taxonomic profilers are affected by read availability in equal measure and other aspects of taxonomic profiling can counterbalance these effects. For example, while Pathoscope 2.0 and Metamix produce some profiles that are less sensitive at sequencing depths of 100K then at sequencing depths of 1M, for Pathoscope 2.0 this only applies for metagenomes containing 100 species. This could be due to the mitigating effect on incorrect read reassignment of having 426 species present in the metagenome.

A strong determining factor in susceptibility to read availability is the use of gene markers instead of full or transformed reference genomes. A combination of low sequencing depth and high species count result in a lower average number of reads per species, regardless of dominance scenario, which limits the overall existing evidence for each species. We observed that under these conditions, the performance of MetaPhlAn2 and ConStrains, is severely degraded. Here the low read count is compounded by the fact that many reads do not overlap with marker genes and are therefore not considered. For a low sequencing depth of 100K, many species will not have sufficient read representation in marker genes to be useful, particularly where some species are represented in low abundance as in an acute infection scenario. As is the case with other taxonomic profilers, MetaPhlAn2 and Constrains work best at high sequencing depths of 10M, where the dearth of reads that match marker genes can be mitigated, but even then, the pipelines' sensitivity is capped by this exclusive use of marker genes over the whole set of available sequences. Marker genes therefore provide an incomplete picture of diverse metagenomes even at high sequencing depths because of the selective use of reads. An incomplete picture can be useful, provided it is not confounded by many false positives, which our study suggests is indeed not the case. However, for profiles of low sequencing depth, marker genes strategies are not nearly sensitive enough to provide useful results. This de facto eliminates marker gene strategies for most in-depth studies. MetaPhlAn2 and Constrains real strength is to be found in their low computational burden, given both have a real computing time much lower than that of taxonomic profilers that rely on whole-genome alignments. However, even this quick turnaround does not always give

marker gene methods an edge on profilers such as Kraken and Centrifuge, which show similar low real computing times.

Unlike other attributes, read length generally has a much smaller and often imperceptible effect over taxonomic profiles. Generally, when taxonomic profiles are less sensitive due to read depth, it will be because of long read depths such as 1,000 bp. This makes sense given that many of the taxonomic profilers were built with short-read sequencing technology, and in some cases, alignment, in mind. The exception is Taxator-tk for which short reads lead to less sensitive taxonomic profiles. This could be due in part to Taxator-tk's reliance on BLAST (*Altschul et al., 1990*), rather than read alignment methods such as Bowtie2 (*Langmead & Salzberg, 2012*), which was designed for short reads. While Metamix also relies on BLAST, its intermediate assembly stage ensures that short reads get turned into longer contigs before being submitted to BLAST, overcoming this issue. It is worth noting that read lengths as small as 75bp are no longer commonly used. Taxator-tk's sensitivity to read length is therefore not particularly disabling in a practical sense and could even prove to be invaluable as sequencing transitions to longer sequencing technologies.

Regardless of which attributes we focus on, some taxonomic profilers are overall more sensitive than others. As mentioned above, marker gene strategies are the least sensitive overall. Among read re-assignment strategies, Sigma and Taxator-tk, while more sensitive than marker gene-based profilers, have the lowest sensitivity in their category. This leaves Pathoscope 2.0, Metamix, along with Centrifuge and Kraken, as the most sensitive taxonomic profilers overall. While Pathoscope 2.0 and Metamix produce the most sensitive taxonomic profiles, they do not perform as well under every scenario. By contrast, while Centrifuge and Kraken have very sensitive, but not the most sensitive, taxonomic profiles, they show very little variability in sensitivity between scenarios. This means that, while Pathoscope 2.0 and Metamix will perform best under certain scenarios, such as high sequencing depth, Kraken and Centrifuge will perform well under every scenario (with the small caveat that Centrifuge has overall less specificity than the other three taxonomic profilers) making them ideal when the parameters of a metagenome are hard to estimate.

Another crucial element that separates out the four highest performing taxonomic profilers is real computing time. Full read alignment and posterior re-alignment requires substantially more computing time than relative to other strategies. Kraken and Centrifuge require real computing times that rival marker gene strategy computing times. In this respect, Centrifuge, largely outperforms other taxonomic profilers. However, shorter real computing time can come at the expense of downstream analysis. It may therefore still be useful to prefer high real computing time in some instances where reference and read data need to be conserved. With overall high sensitivity, Pathoscope 2.0, Metamix, Kraken and Centrifuge are all useful given different research objectives. Gene marker strategies, such as MetaPhlan2 and Constrains, while still useful for a quick characterization of highly abundant species, given high depth, are destined to be completely phased out as taxonomic profilers as alternative approaches such as Centrifuge compete with these in real computing time.

## CONCLUSIONS

By simulating 144 possible metagenomics scenarios for eight taxonomic profiling pipelines, we produced a total of 1,104 taxonomic profiles from which sensitivity and specificity metrics could be calculated. Coupled with other considerations such as standard error, genome absence and running time, we were able to describe how different taxonomic profilers fare faced with data showing different attributes. For all taxonomic profilers, species in very low abundance, as when one species is present in half of the reads, lowered sensitivity across the board, other profile attributes held equal. Taxonomic profilers using gene markers such as MetaPhlAn2 and Constrains were particularly negatively impacted by both low abundance species specifically and lower sequencing depth generally. Both tools allow relatively fast assessment of the most abundant species, making them still useful in obtaining a quick overview of a metagenome, provided several species are present in high abundance, as in the null case. It is therefore particularly useful in the event of an infection, the infectious species overshadowing all others.

Tools and pipelines that relied on alignment to full sequences, fared differently based on a combination of alignment source, BLAST or Bowtie2, and the read reassignment method. In many respects, taxonomic profilers are susceptible to different attributes with varying intensities, but this was most obvious for read reassignment methods. Independent of the variability in sensitivity and specificity within pipelines, taxonomic profilers using a Bayesian approach to read reassignment, such as PathoScope 2.0 and MetaMix, overall produced better results. However, these tools also required the most computational time. For studies where computing time is less of a concern, but the highest sensitivity is a priority, as in a detailed study characterizing the microbiome of a given organ, these tools should be prioritized. Studies that require more downstream analysis, such as strain variant detection, can also benefit from a profiler such as PathoScope 2.0 which conserves alignments after read reassignment.

Other tools that derive their strength from compression of whole genomes can also provide much more complete taxonomic profiling that capture lower abundance species with similar if not faster computing times. This is the case with tools such as Kraken and Centrifuge. Studies that require a high level of accuracy, but also remain time-sensitive, would benefit most from these tools. Health studies that include very large quantities of case and control metagenomes would fit this category.

While our analysis certainly highlights the excellence of several of the more recent taxonomic profilers, it also shows that no one taxonomic profiler yet possesses every single quality that might make it an undisputed gold standard. This is why evaluations of methods, such as the one above, are important, not only to assess which tools provide some of the best results, but also under which conditions one particular method will do better than another. In order to benefit from the various strengths of these taxonomic profilers, researchers may consider using more than one taxonomic profiler for studies attempting to answer a scientific question in several steps. For example, marker gene profiling could be used as a preliminary for selecting metagenomes to study more deeply using tools like PathoScope 2.0. It is important to note however that future methods might

render this combination of methods unnecessary, as the speed and accuracy of methods such as Kraken suggests. As the databases these tools rely on are continually upgraded and new methods are developed, it is important to be able to keep evaluating these methods with those already established. For this reason, all the data from this analysis are available for further evaluation via the R Shiny application set up for this purpose. This study therefore serves as an initial comparison to which new or upgraded pipelines can be added as these are released.

## ACKNOWLEDGEMENTS

We would like to thank the high-performance computing facility from The George Washington University, ColonialOne, for providing data storage, support, and computing power for all the analyses described in this article.

### Funding

This project was supported by Award Number UL1TR001876 from the NIH National Center for Advancing Translational Sciences. It was also supported by ANID-PAI 82140008, ANID-FONDECYT Regular 1200834, and ANID-PIA-Anillo INACH ACT192057. The funders had no role in study design, data collection and analysis, decision to publish, or preparation of the manuscript.

### Grant Disclosures

The following grant information was disclosed by the authors:
NIH National Center for Advancing Translational Sciences: UL1TR001876.
ANID-PAI: 82140008.
ANID-FONDECYT: 1200834.
ANID-PIA-Anillo: ACT192057.

### Competing Interests

Keith A. Crandall and Eduardo Castro-Nallar are section and academic editors for PeerJ, respectively.

### Author Contributions

- Matthieu J. Miossec performed the experiments, analyzed the data, prepared figures and/or tables, authored or reviewed drafts of the paper, and approved the final draft.
- Sandro L. Valenzuela performed the experiments, analyzed the data, prepared figures and/or tables, and approved the final draft.
- Marcos Pérez-Losada conceived and designed the experiments, authored or reviewed drafts of the paper, and approved the final draft.
- W. Evan Johnson conceived and designed the experiments, authored or reviewed drafts of the paper, and approved the final draft.
- Keith A. Crandall conceived and designed the experiments, authored or reviewed drafts of the paper, and approved the final draft.

- Eduardo Castro-Nallar conceived and designed the experiments, performed the experiments, analyzed the data, prepared figures and/or tables, authored or reviewed drafts of the paper, and approved the final draft.

## Data Availability

Data, code, and visualization through dedicated Shiny App available at GitHub: https://github.com/microgenomics/HumanMicrobiomeAnalysis.

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
