# Peer review of "Evaluation of computational methods for human microbiome analysis using simulated data"

_PeerJ, doi:10.7717/peerj.9688_

## Round 0.1 · original submission · Major Revisions

All 3 reviewers find your manuscript valuable, but point out that a comparison to real data is missing and would be a valuable test of your performance in less-than-ideal conditions. They also have several additional suggestions to further improve the clarity of the manuscript.

·

Basic reporting

Studies comparing metagenomic algorithms are useful to both experts in the field, and researchers who are just considering applying metagenomics. Therefore, to help both groups, I'd suggest the following changes:
- Discussing beefily the difference/similarities between binning and profiling approaches to metagenomic analysis.
- Line 152: DIAMOND can align a large number of reads to the large databases, thanks to the spaced-seed. Buchfink, B., Xie, C., & Huson, D. H. (2014). Fast and sensitive protein alignment using DIAMOND. Nature Methods, 12(November), 59–60. https://doi.org/10.1038/nmeth.3176.
- You mention the diversity of the dataset as an important factor impacting performance of the analysis pipelines. However, it would be good to discuss the concept of diversity mentioning the relative abundance of the species.
- I’m not too familiar with the Oral microbiome. It would be nice if you could add a couple of sentences describing the general structure of the oral microbiome, how many phyla? How much inter-personal diversity etc.
- The authors explained well that the profiling algorithms can only be as good as robust and comprehensive as is the underlying reference database
- The authors describe in detail the differences of the selected pipelines, but I believe a graphical representation would better serve as an overview and comparison of the analysis pipelines.
- In conclusions: you could attempt to give a simple recipe to the metagenomics newbies, which tool to use regarding the data, available time, computational power and diversity.
- In your opinion, could there be value in using some combination of tools?

Experimental design

- Why did you pick 75-100 bp as the read length? This way you are leaving out the popular sequencing technology such as Illumina MiSeq with paired-ends.
- You also have not mentioned the long reads technologies like PacBio or Oxford NanoPore. Although their application to metagenomics is still in the beginning of the development, I think you should better justify the read length.
- Kraken is being rapidly developed, I think using the latest versions of the software (Kraken2) would be much more useful.
- Using perfect data constitutes a good baseline to pick the best suited tool for the problem. But this way you do not compare how robust the tools are regarding the error rate, which is an important factor.

Validity of the findings

- The authors thoroughly analyze the results and pose well justified conclusions.
- Graphical representation of the simulated profiles (e.g. a stack chart) and read length would be quite informative.
-Using the newest Kraken, adding longer reads, adding non-perfect reads would much improve this publication.

Additional comments

a. Lines 201 – 208 I’d add the references here.
b. Figure 2: It is hard to distinguish the shapes and colors.
c. Figure 3: if the x-axis represents the different pipelines, what does it mean if the data-point is located between two ticks on x-axis?
d. Figure 4: I suggest to flip this plot, namely to show how does the performance of pipeline change with the diversity.

Reviewer 2 ·

Basic reporting

Clear, unambiguous professional language used throughout.

For the most part, clear language is used. However, there were a few noted instances of unclear language:

Line 84 – ‘of one sort or another.’
Line 397 – ‘nearly the entire gamut’
Line 402 – ‘break this trend’
Line 425 – ‘vary very little’
Line 694 – ‘buck that trend’
Line 660 – ‘occupies a space in between’

The authors may benefit from reading through the manuscript and clarifying language for an easier read.

Intro and background to show context.

The introduction does a great job showing the progression of taxonomic classifier strategies over time, effectively showcasing the differences among classifiers and their strengths/ weaknesses. Also, presenting the manuscript as a strategy for future comparisons as more classifiers are released shows the sustained impact this manuscript can have.

However, some information presented in the methods would benefit to clarify the descriptive information about how the classifiers work as added detail in the introduction. Classifier background information could be integrated in the paragraph on line 145. For more information, please see Experiment Design: Methods described with sufficient detail and information to replicate portion of the review.

Also, the background about different methods of creating complete metagenomic reference sets on line 104 could be further synthesized to the most relevant information. This paragraph and subsequent small paragraphs (starting at line 122 and line 128) can be combined and synthesized as well.

Structure conforms to PeerJ standards, discipline norm, or improved for clarity.

The results and discussion section were combined, and it would benefit the manuscript to separate them. Also, a separate funding statement is required outside of the acknowledgements.

Figures are relevant, high quality, well labeled, and described.

Figure 1 is not necessary if a link to the GitHub page is included.

Figure 2 text is too small to read, specifically the axis and legend. The dot size can be increased, and transparency of the dots reduced. For the bottom two panels especially, can the y-axis range be reduced (noting that in the figure caption) so it shows a clearer picture.

Figure 3 – The title is unnecessary. Also, can the authors need to clarify how outliers were determined in the figure caption.

Figure 4 – The text and data points should be a little larger.

Figure 5 – Axes and legend labels should have larger text.

Raw data was supplied.

Raw data were supplied on Github. Data provided are sufficient to replicate results.

Experimental design

Original primary research within the Scope of the journal.

Primary research was within the scope of PeerJ. This manuscript has relevance to biological, environmental, and medical sciences as selecting tools to characterize the microbiome have broad reaching impact. This research article is scientifically sound and identifies an important knowledge gap relevant to many fields.

Research question well defined, relevant, and meaningful. It is stated how the research fills an identified knowledge gap.

Research fits the aforementioned criteria. It fills a research gaps as there is limited research on the topic, however the results are useful to many different fields.

Rigorous investigation performed to a high technical and ethical standard.
The authors did a very thorough job documenting how the taxonomic profilers differ and comparing their performances. They included several different parameters for a vigorous comparison. Technically, the methods are sound and effective. There are no ethical concerns with this study.

Methods described with sufficient detail and information to replicate.

Overall, the methods described are detailed enough for replication. The authors should include code for running each taxonomic classifier on the GitHub page, to make the work a more useful resource for readers.

The methods overall were well justified and rigorous. Microbial/metagenomic attributes were informed by biological conditions, as well as read length and sequence depth by current technologies. Including the equations for RRMSE and AVGRE were useful for reader understanding and replication. The logic presented on excluding certain pipelines for downstream analyses was sound. Adding an unknown taxon and determining the effect on relative abundances has real-world significance, as well as addressing run time, which is often neglected in bioinformatic studies.

However, descriptions of the taxonomic profilers in the materials and methods read like background information. Some of this information might be more effective in the introduction or synthesized into the most important information in material and methods. For example, lines 218-221 and lines 223-230 could be further synthesized down to the most important information. Emphasizing the justification as to why each taxonomic profiler was chosen would make that section more useful for readers.

Line 312 – Because you have one biological scenario to mimic microbial data, i.e. acute infection, are there scenarios that reflect the small group of species show high abundance and half of species account for a majority of reads?

Line 331- why was +/- 50% of the actual abundance chosen as a match?

Validity of the findings

Impact and novelty not assessed. Negative/inconclusive results accepted. Meaningful replication encouraged where rationale and benefit to literature is clearly stated.

The structure of the results/discussion is effective. However, further rational and benefit to literature could be explored in the result/discussion section, as a lot of data were presented that could benefit from further discussion/summarization.

All underlying data have been provided; they are robust, statistically sound, and controlled.

Underlying data meet these criteria. There are a lot of data presented, and it would benefit the reader to summarize information and present main take home points.

There are not a lot of statistical tests are included in the manuscript, but sensitivity and specificity among classifiers with mean and variances test?

Speculation is welcome, but should be identified as such.

Speculation is not a large portion of this manuscript.

Conclusions are well stated, linked to original research question and limited to supporting results.

The conclusions are a little unclear, but could be due to the combined results/discussion format. However, conclusions are linked to the original research question and presented results.

Additional discussion points could improve the overall manuscript:

As the taxonomic profilers were compared under perfect base calling conditions, it might be useful to add to the discussion a comment about how the profilers would be affected by real data with sequencing errors.

A more clear summary of the tradeoffs among the pipelines and under which scientific scenarios would be best for each taxonomic classifier (presented in a table might be most effective) is needed. Also, based on the taxonomic profiles strengths/weakness there should be a better link back to the biological significance presented earlier in the manuscript (e.g., MetaPh1An2 would not be useful for disease states).

While the null state (presence of species in roughly equal number) was tested, the authors should add more discussion about what the results mean for the null case?

Additional comments

This manuscript has a lot of potential and will be an excellent contribution to high throughput sequencing methods; however, the overall manuscript is dense and long, and could be streamlined (especially in the methods and results/discussion) to make a better, tractable read for a more general scientific audience.

Line 210 – splitting up this paragraph when MetaPhlAn is introduced (line 218) will add clarity.

Line 328 – Definitions of sensitivity and specificity could be more clearly explained.

Line 497 – ‘little to be said’ sentence is unnecessary

Reviewer 3 ·

Basic reporting

Given the increasing number of metagenomic analysis tools, it is very important to have a comprehensive evaluation/comparison of different tools. Thus I believe that this work is important and timely.
Overall, the writing is good but some parts are hard to follow, which can be particularly hard for readers who are not familiar with the methods. Several parts read like a laundry list. Better organizations should be applied to highlight the most important findings.

Experimental design

The authors did experiments using a large number of simulated metagenomic data and thus the results should be helpful for users to choose an appropriate tool. But one concern is that all the data are simulated. Although using simulated data can help quantify the performance more accurately, lacking results on real data is worrisome. Usually, the performance on real data and simulated data can be different, sometimes much worse. And this is not only because of the sequencing errors.

Can the authors compare these tools on one or two real data sets? Just comparing the findings among different tools will provide important information, if lacking ground truth is a concern.

Some tools were designed for slightly different purposes. For example, a couple of them can conduct strain-level analysis while others cannot. So, in this case, should the utilities specific to each tool be considered in the comparison? Or at least provide a table summarizing that information for users?

Validity of the findings

No comment

Additional comments

Some detailed comments are listed here:
1. I think it is not accurate to call Bowtie a heuristic. When conducting exact match, BWT-based tools can return correct matching positions. When errors are allowed, Bowtie may only return some of the matched positions. But overall, its main indexing structure (BWT) is not a heuristic.
2. The authors chose Bowtie but did not give justifications. Some people prefer BWA. They do have different outputs under their default parameters (such as local vs global). The authors need to discuss this because using different read mapping tools can lead to different downstream analysis output.
3. Line 331, any justification for choosing 50%? This sounds like a very permissive threshold.

---

## Round 0.2 · Minor Revisions

Please address the remaining minor points

·

Basic reporting

It is a second-round review, therefore I have only a couple of general remarks:

I believe the reviewed publication in naming and explaining the challenges in the short-read metagenomics, and is going to benefit scientists both new to metagenomics, thanks to the extensive introduction), and experts seeking the best tool. A continuously updated benchmarking tool is an important innovation. I hope authors consider extending their benchmarking efforts on the one side to 16S rRNA sequencing studies, and on the other side to the long-reads analysis.

Experimental design

The experimental design fits the question - the benchmarking of the pipelines from taxonomic profiling of the whole-genome metagenomic studies. The authors justified their choice of the:
- read length
- error rate (the reads are perfect because they wanted to compare the pipelines regardless of the error rates)
- the taxonomic composition (mock human oral microbiome)

Validity of the findings

Hope the authors commit to maintaining the benchmarking git repository,

Additional comments

Some language issues that I believe could be clarified that I managed to find:
- Line 86: I think “in turn” misleads the reader in this context, I would suggest a rewrite.

With taxonomic binning, individual sequence reads are clustered into new or existing operational taxonomic units (OTUs), obtained through sequence similarity and other intrinsic shared features present in reads. In turn, taxonomic binning can contribute to taxonomic profiling. With taxonomic profiling, the focus is on estimating the presence and quantity of taxa in a microbial population as well as the relative abundance of each species present.

- Line 153: consisted in reducing a multiple sequence
- Line 167: Other taxonomic profilers have therefore been developed around the desire to use entire genome libraries for more accurate taxonomic profiling and better functional inference, Is too long and convoluted.
- Line 181: .. before classification can take place.

---

## Round 0.3 · accepted · Accept

I am glad to accept your paper, which I believe will have much impact in the field